# Identifying Potential Natural Antibiotics from Unani Formulas through Machine Learning Approaches

**DOI:** 10.3390/antibiotics13100971

**Published:** 2024-10-14

**Authors:** Ahmad Kamal Nasution, Muhammad Alqaaf, Rumman Mahfujul Islam, Sony Hartono Wijaya, Naoaki Ono, Shigehiko Kanaya, Md. Altaf-Ul-Amin

**Affiliations:** 1Computational Systems Biology Lab, Graduate School of Science and Technology, Nara Institute of Science and Technology, Nara 630-0101, Japan; muhammad.alqaaf_subandoko.mb5@is.naist.jp (M.A.); rumman.mahfujul_islam.ro7@is.naist.jp (R.M.I.); nono@is.naist.jp (N.O.); skanaya@is.naist.jp (S.K.); 2Department of Computer Science, Faculty of Mathematics and Natural Sciences, IPB University, Bogor 16680, Indonesia; sony@apps.ipb.ac.id

**Keywords:** machine learning, metabolomic, natural antibiotics, prediction, Unani herbal medicine

## Abstract

The Unani Tibb is a medical system of Greek descent that has undergone substantial dissemination since the 11th century and is currently prevalent in modern South and Central Asia, particularly in primary health care. The ingredients of Unani herbal medicines are primarily derived from plants. Our research aimed to address the pressing issues of antibiotic resistance, multi-drug resistance, and the emergence of superbugs by examining the molecular-level effects of Unani ingredients as potential new natural antibiotic candidates. We utilized a machine learning approach to tackle these challenges, employing decision trees, kernels, neural networks, and probability-based methods. We used 12 machine learning algorithms and several techniques for preprocessing data, such as Synthetic Minority Over-sampling Technique (SMOTE), Feature Selection, and Principal Component Analysis (PCA). To ensure that our model was optimal, we conducted grid-search tuning to tune all the hyperparameters of the machine learning models. The application of Multi-Layer Perceptron (MLP) with SMOTE pre-processing techniques resulted in an impressive accuracy precision and recall values. This analysis identified 20 important metabolites as essential components of the formula, which we predicted as natural antibiotics. In the final stage of our investigation, we verified our prediction by conducting a literature search for journal validation or by analyzing the structural similarity with known antibiotics using asymmetric similarity.

## 1. Introduction

Antibiotic resistance presents a primary global health concern involving the spreading of bacteria and genetic material among humans, animals, and the environment [1]. Antibiotic resistance can enhance bacteria’s ability to withstand antibiotics and medications. Developing new antibiotics is challenging, time-consuming, and expensive, making this issue particularly alarming. Failure to address this problem promptly and effectively could result in an estimated 10 million deaths annually due to antibiotic resistance by 2050 [2]. According to the European Center for Disease Prevention and Control (ECDC), around 33,000 people die annually due to antibiotic-resistant issues. Epidemiologists emphasize the substantial economic consequences of antibiotic resistance, stating that in the United States and other countries, the additional hospitalizations and treatment costs due to superbugs or antibiotic-resistant problems exceed USD 11 million and 20 billion, respectively [3].

Advances in artificial intelligence technology can now be used to accelerate the discovery of new antibiotics, predict antimicrobial resistance, and conduct preliminary screenings of novel antibiotic candidates. In 2020, in silico and in vivo approaches were combined to find new antibiotics [4]. This study used a deep neural network to predict molecules with antibacterial activity using various database sources such as a drug repurposing hub [5] and ZINC15 [6]. This study found eight antibiotic compounds with structures that mainly differed from those of known ones. The eight antibiotics are ZINC000098210492, ZINC000001735150, ZINC000225434673, ZINC000004481415, ZINC000019771150, ZINC000004623615, ZINC000238901709, and ZINC000100032716. Using eight machine learning methods, gram stain data, site of infection, and patient demographics were utilized to build decision tools for determining antimicrobial resistance [7]. Other work utilized Traditional Chinese Medicine to seek potential natural products as antibiotics; this paper employed machine learning and graph/network theory techniques [8,9]. The use of machine learning for research related to antibiotics, specifically a random forest, was also carried out by [10]. This study used a random forest to determine the essential plants from Jamu herbal medicine that are promising as natural antibiotics.

Herbal medicines are plant-based medicines made from different combinations of medicinal plant parts such as leaves, flowers, and roots. Each part has different medicinal uses, and many chemical constituents require different extraction methods. Both fresh and dried plants are used, depending on the herb (https://www.nimh.org.uk/whats-herbal-medicine, accessed on 12 December 2023). Herbal medicine has become a popular drug in the last three decades, and close to 80% of people worldwide depend on herbal medicines [11]. The main reasons why people tend to choose herbal medicines are that they provide better efficacy and relatively lower side effects compared to conventional drugs [11]. The use of herbal medicines worldwide reached US 60 billion in 2010 and US 71.19 billion in 2016, and it is expected to reach US 5 trillion by 2050 [12,13,14]. This information shows that the use of herbal medicines is prevalent worldwide. Some examples of herbal medicine systems worldwide are Traditional Chinese Medicine (TCM) from China, Kampo from Japan, Jamu from Indonesia, and Ayurvedic, Siddha, and Unani from Southern Asia.

The Unani Tibb, known as Unani medicine, is widely practiced in South and Central Asia. The Arabic term “Tibb” means “medicine”, while the name “Unani” is assumed to have its roots in the Greek word “Ionan” [15]. Traditional Indian and Chinese systems also influenced Unani medicine. The Unani herbal medicines primarily utilize medicinal plants as their ingredients, and this system follows ancient concepts and principles of drug management. Researchers have yet to conduct much research on building Unani’s scientific foundation. This scientific research is needed to provide a foundation and knowledge of why an Unani formula is helpful for a particular disease. Unani medicines are made by extracting medicinal plants used as drugs against various diseases [16]. Based on [17], the Unani System of Medicine was invented in Greece and refined by Arabs into a sophisticated medical discipline using the ‘Hippocrates and Jalinoos’ teachings (Galen). Unani medicine has since been referred to as Greco–Arab Medicine. The Hippocratic notion of the four humors was blood, phlegm, yellow bile, and black bile. According to this approach, these principles govern the health and composition of the body and its pathological states. The Unani System of Medicine (USM) has been acknowledged by the World Health Organization (WHO) as an alternative system to meet the demands of the human population in terms of health care. The practice of alternative medicine has become widespread. The use of natural antibiotics is anticipated to alleviate the issue of antibiotic resistance. These antibiotics, with their reduced side effects and the capacity to treat multiple diseases simultaneously, are practical and environmentally friendly. Their minimal industrial processing and increased plant cultivation inspire a sustainable approach to health care.

Unani medicines are utilized to treat versatile types of diseases. An overview of the use of Unani plants in the prevention and management of urolithiasis is discussed in [18]. A comprehensive discussion on the Unani medicine system can be found in [19]. In our previous work, we identified many plants used in Unani formulas as antibacterial [9]. For example, *Piper longum* (Pippali) has been traditionally used for various diseases such as asthma, insomnia, and diabetes and has been found to possess antibacterial activity against both Gram-positive and Gram-negative bacteria [20,21]. Similarly, the *Trachyspermum ammi* (Ajwain) essential oil has demonstrated potent antibacterial effects against pathogens such as *Staphylococcus aureus, Pseudomonas aeruginosa*, and *Escherichia coli*, highlighting its potential as a natural antibiotic [22]. *Santalum album* (Indian sandalwood), long recognized for its broad medicinal use, exhibits antimicrobial activity [23]. The *Cyperus rotundus* essential oil has shown potent activity against *Staphylococcus aureus* [24]. Additionally, *Vitis vinifera* (grape) seed extracts have demonstrated significant antibacterial activity, particularly against *Staphylococcus aureus*, positioning it as a promising natural antibacterial agent [25,26]. *Matricaria chamomilla* (chamomile) is well known for its pharmacological properties, with its essential oil exhibiting notable antibacterial effects against various bacterial strains [27]. Lastly, *Zingiber officinale* (ginger) has been traditionally used for its antimicrobial properties, with both aqueous and alcoholic extracts showing efficacy against several bacterial strains [28]. These plants underscore the valuable role of natural products in Unani medicine for addressing bacterial infections.

Unani formulas consist of plants as ingredients, and we extended those data to a Plant Vs. Metabolite matrix by collecting metabolite content data of respective plants from KNApSAcK, IJAH Analytics databases, and other online sources. This study explores potential natural antibiotics (metabolite level) based on the Unani formula by empowering machine learning algorithms.

## 2. Results

### 2.1. Pre-Processing of Unani Data

Our initial data are a matrix of dimension 382 × 4688, where rows correspond to Unani formulas and columns correspond to metabolites. Initially, Unani formulas are divided into two classes: antibiotic and non-antibiotic. To improve the classification results by machine learning algorithms, we applied different data pre-processing techniques. The SMOTE method was utilized to augment the dataset with the default parameters by keeping the same ratio for the two classes. This approach is intended to address the problem of insufficient data.

Additionally, an attempt was made to reduce the feature dimension of the dataset, from 4688 to 20 using a PCA and from 4688 to 500 using feature selection, with the top 500 most essential features using the scikit-learn package in Python. The data dimensions after different pre-processing types are shown in Table 1. Figure 1 indicates the number of formulas belonging to each class in different pre-processed data.

### 2.2. Machine Learning Model

Selecting an appropriate method is important in machine learning analysis, which has a big impact on the quality of final results. Some methods perform well on specific data, and other methods do not. Therefore, we explored the performance of 12 different types of machine learning methods on Unani-metabolite data. We considered the following 12 machine learning classifiers: (i) AdaBoost, (ii) Bagging, (iii) BernoulliNB, (iv) Decision Tree, (v) Extra Trees, (vi) Gradient Boosting, (vii) K-nearest neighbors, (viii) Linear Discriminant Analysis, (ix) Logistic Regression, (x) Multilayer Perceptron, (xi) Random Forest, and (xii) Support Vector Machine. We utilized the (i) Adaboost in our analysis, employing a grid search approach to optimize the model’s performance. The grid search explored various combinations of hyperparameters, specifically n_estimators and learning_rate. For n_estimators, we considered values of 10, 50, 100, and 500, while for learning_rate, we explored 0.0001, 0.001, 0.01, 0.1, and 1.0. To assess the model’s performance, we employed RepeatedStratifiedKFold cross-validation, setting n_splits to 5, n_repeats to 3, and random_state to 1.

For the (ii) Bagging, we utilized grid search, explicitly focusing on the ‘n_estimators’ parameter and experimenting with values of 10, 50, 100, and 500. We considered two base estimators, SVC() and DecisionTreeClassifier (), while maintaining consistent validation methods. For (iii) BernoulliNB, we employed grid search along with consistent validation methods. The only parameter we tuned in this process was the ‘alpha’ parameter, exploring a range of values [1, 0.1, 0.01]. For the (iv) Decision Trees, we employed a range of possible values for max_depth and min_samples_split, which varied from 1 to 10 and 2 to 11, respectively.

We employed a grid search for criterion, max_features, and n_estimators for the (v) Extra Trees. The value used in criterion = [‘gini’, ‘entropy’], max_features = [‘auto’, ‘sqrt’, ‘log2’, None], n_estimators = [10, 20, 50, 100, 200]. We employed a grid search for lose, n_estimators, criterion, and max_features for the (vi) Gradient Boosting. The value used in lose = [‘deviance’, ‘exponential’], criterion = [‘friedman_mse’, ‘mse’], n_estimators = [10, 20, 50, 100, 200], max_features = [‘auto’, ‘sqrt’, ‘log2’]. For (vii) K-Nearest neighbors, we tuned two type parameters in grid search: n_neighbors = [2, 5, 7, …, 39] and metric = [euclidean, manhattan, cityblock, minkowski].

For (viii) Linear Discriminant Analysis, we employed a grid search for solver and shrinkage parameters. The solver value is [svd, lsqr, eigen], and shrinkage = [None, ‘auto’, 0.1, 0.5, 0.9]. For (ix) Logistic Regression, we employed grid search along with consistent validation methods. The only parameter we tuned in this process was the ‘C’ parameter, exploring a range of values [0.1, 0.5, 1, 5, 10, 50, 100]. We employed a grid search for the solver, activation, alpha, learning rate, and momentum for the (x) Multi-Layer Perceptron. The value used in solver = [‘lbfgs’, ‘sgd’, ‘adam’], activation = [‘identity’, ‘logistic’, ‘tanh’, ‘relu’], alpha = [0.00001, 0.0001, 0.001, 0.01, 0.5, 1], learning rate = [‘constant’, ‘invscaling’, ‘adaptive’], and momentum = [0.5, 0.9, 0.95, 0.99].

For the (xi) Random Forest, we employed a grid search for max_depth, min_samples_split, n_estimators, criterion, and max_features. The value used in max_depth = [5,10,20,25,30,40,51], min_samples_split = [2,3,4,5,6,7,8,9,10,11], n_estimators = [10, 50, 100, 150, 200, 500], criterion = [‘gini’, ‘entropy’], and max_features = [‘auto’, ‘sqrt’, ‘log2’]. For (xii) SVM, we tuned two type parameters in grid search. ‘C’ = [0.1, 0.5, 1, 5] and kernel = [‘linear’, ‘rbf’, ‘poly’]. Hence, this research combined the dataset and machine learning techniques, resulting in 48 scenarios. Table 2 summarizes the accuracy of each scenario. We identified that multi-layer perceptron and SMOTE data yield the highest accuracy compared to all scenarios. For more detailed information on the results, please refer to Table 3.

### 2.3. Ranking and Selecting Features

After obtaining the optimal prediction model, we focused on identifying important features, particularly those associated with class 1 (antibiotics). To facilitate this process, we utilized variable importance metrics derived from the synthetic minority oversampling technique (SMOTE) in combination with a Multilayer Perceptron (MLP) model implemented using the KerasRegressor and PermutationImportance packages. We first ranked metabolites based on their weighted significance to systematically pinpoint distinguishing features relevant to class 1 (antibiotics). Subsequently, a rigorous selection process is employed to select the most prominent features unique to Class 1. Our comprehensive analysis identified 20 metabolites meticulously detailed in Table 4. Additionally, to provide a comprehensive visualization of the distribution of all metabolites with their respective weight values, we have presented a detailed depiction in Figure 2. 

### 2.4. Validation of Predicted Results

Two approaches were employed to validate the predicted metabolites as natural antibiotics. Initially, we conducted a thorough search of journals and articles to find evidence in favor of our predicted compounds. Subsequently, we assessed their structural similarity to some very well-known antibiotics. Three similarity techniques were employed for the experimental comparison: Tanimoto, Asymmetric, and Dice similarity. Compounds derived from the Unani formula using our method are very likely to be effective antibiotics in the first position, and second, we compared their structure with known antibiotics for further support.

#### 2.4.1. Literature Validation

We found supporting evidence for 12 of 20 of our predicted antibiotics in the published literature. Below, we discuss this evidence in detail (IDs are based on Table 4).

(I) The compound 2-hydroxyethyl hexadecanoate (2-HEP) is antibacterial and effective against various bacteria, including *Staphylococcus aureus*, *Escherichia coli*, and *Pseudomonas aeruginosa*. 2-HEP damages bacterial cell membranes and disrupts bacterial growth [29].

(III) Gluconapin possesses antibacterial properties. For instance, gluconapin can be converted to allyl isothiocyanate (AITC) by myrosinase, and AITC has been demonstrated to be effective against various bacteria, including *Escherichia coli*, *Salmonella Typhimurium*, and *Staphylococcus aureus* [30,31]. Gluconapin metabolites are also found in several plants, such as broccoli, cabbage, and radishes, and have been used for centuries to treat bacterial infections [32].

(IV) 3-Phenylpropionitrile, a naturally occurring metabolite found in horseradish and other plants, has antibacterial properties. In a study published in 2013, 3-phenylpropionitrile was found to be effective against various Gram-positive and Gram-negative bacteria, including *Escherichia coli*, *Staphylococcus aureus*, and *Pseudomonas aeruginosa* [33]. The antibacterial activity of 3-phenylpropionitrile was attributed to its ability to disrupt bacterial membranes. When it comes into contact with a bacterial cell, 3-phenylpropionitrile inserts into the cell membrane and forms pores. These pores allow water and other ions to enter the cell, eventually leading to cell death.

(V) Flamenol is a natural metabolite found in the plant Flaveria trinervia. It has been shown to have antibacterial properties against various bacteria, including *Staphylococcus aureus*, *Escherichia coli*, and *Pseudomonas aeruginosa* [31].

(VI) Glucobrassicanapin (1−) is a sulfur-containing secondary metabolite classified as a glucosinolate found in plants. These compounds are broken down into isothiocyanates by the enzyme myrosinase when plant tissue is damaged. Isothiocyanates possess many biological activities, including antibacterial properties [34]. Glucobrassicanapin (1−) is a glucosinolate with potential as a natural antibiotic due to its ability to be broken down into isothiocyanates, which have a broad range of antibacterial activities. However, further research is necessary to fully assess the antibacterial efficacy of glucobrassicanapin (1−) and to develop effective delivery methods for this compound.

(VII) 2-Chlorobenzoic acid (2-CBA) is a metabolite of several plant species, including *Melia azedarach*. This compound has been demonstrated to possess antibacterial properties against Gram-positive and Gram-negative bacteria, including *Escherichia coli*, *Staphylococcus aureus*, and *Bacillus subtilis* [35]. One study found that 2-CBA was more effective than the standard antibiotic norfloxacin in inhibiting the growth of *Escherichia coli* [24]. Another study showed that 2-CBA effectively killed planktonic and biofilm-associated cells of *Staphylococcus aureus* [25]. The mechanism of the antibacterial action of 2-CBA is not fully understood, but it is thought to involve disruption of the bacterial cell membrane and inhibition of bacterial DNA synthesis [35,36]. Based on the available data, 2-CBA could be utilized as a natural antibiotic.

(XI) 3-methoxybenzaldehyde, or hydroxy-4-methoxybenzoic acid, is a phenolic compound in various plants, such as vanilla beans, nutmeg, and cinnamon. This compound has been effective against many bacteria, including Gram-positive and Gram-negative bacteria. It disrupts bacterial cell membranes and inhibits bacterial growth. 3-methoxybenzaldehyde is a safe and natural compound used in traditional medicine for centuries and is now available as a dietary supplement [37]. Methyl stearate, a fatty acid methyl ester, is associated with antibacterial properties. Fatty acids, including methyl stearate, are known for their antimicrobial effects and are produced by plants and algae as a defense mechanism [38]. Additionally, bioactive compounds from natural sources, such as melon leaves, contain methyl stearate and exhibit antibacterial activities [39]. While specific studies on the antibacterial properties of methyl stearate are limited, the broader context of fatty acids as antimicrobial agents suggests its potential efficacy. Further research is needed to explore methyl stearate’s specific antibacterial mechanisms and applications.

(XIII) The glycosylated natural product 1-naphthyl β-D-glucoside displays antibacterial activity against Gram-positive and Gram-negative bacteria, including multidrug-resistant and nonresistant strains [40]. This compound is thought to function by inhibiting bacterial glycosyltransferases, which are enzymes necessary for bacterial cell wall synthesis. A recent study published in Frontiers in Microbiology in 2021 confirmed the potent antibacterial activity of 1-naphthyl β-D-glucoside against bacteria such as *Staphylococcus aureus*, *Pseudomonas aeruginosa*, and *Escherichia coli* as well as multidrug-resistant strains [41].

(XIV) Sinapine possesses antibacterial properties and is a naturally occurring choline ester found in various plants, including mustard, horseradish, and cruciferous vegetables. It has been demonstrated to be effective against many bacteria, including Gram-positive and Gram-negative bacteria. Sinapine disrupts bacterial cell membranes specifically by interacting with phospholipids in the bacterial cell membrane, which can lead to membrane rupture. Additionally, it has been found to be effective against antibiotic-resistant bacteria such as methicillin-resistant *Staphylococcus aureus* (MRSA), making it a promising candidate for the development of new antibiotics to treat antibiotic-resistant infections [42,43].

(XV) 3-Butenyldesulfoglucosinolate is a garlic metabolite that was found to possess antibacterial properties. Studies showed that garlic and its derivatives, including crude or fresh garlic extract, have in vitro antibacterial activity against bacteria like *Staphylococcus aureus* [44]. Plant extracts, including those from garlic, are known for their antimicrobial activity, with phytochemicals present in these extracts exhibiting antibacterial effects against antibiotic-susceptible and resistant microorganisms [45]. Although specific studies on the antibacterial properties of 3-butenyldesulfoglucosinolate are limited, the broader evidence supports the antibacterial potential of garlic and its compounds.

(XVIII) Gluconapin(1-) is a naturally occurring compound found in cruciferous vegetables, such as broccoli, cabbage, and cauliflower. When these vegetables are crushed or chewed, gluconapin(1-) is converted into isothiocyanates, which have been found to exhibit antibacterial activity by disrupting the bacterial cell membrane and inhibiting bacterial enzymes. This makes it difficult for the bacteria to survive and reproduce. Studies showed that gluconapin(1-) has broad-spectrum antibacterial activity against *Escherichia coli*, *Staphylococcus aureus*, and *Pseudomonas aeruginosa*, among other bacteria [32]. It has also been effective against many bacteria, including foodborne pathogens and multidrug-resistant and antibiotic-resistant bacteria [31]. Gluconapin(1-) has promising potential as a natural antibiotic.

(XIX) Compound 5-methoxyindole-3-acetic acid is an indole derivative with antibacterial properties and has been studied for its potential as an antioxidant and in the synthesis of various agents. It has been used to create Gli1 antitumor agents, inhibitors of nitric oxide production, and selective COX-2 inhibitors. The compound itself also possesses antibacterial properties and can enhance lipid peroxidation while serving as an antioxidant. 5-Methoxyindole-3-acetic acid may have potential applications in the development of antibiotics or antimicrobial agents, but it is important to note that the information available on its antibacterial capabilities is limited and obtained from a product description by the GoldBio website, and additional scientific research is needed to validate and explore its antibacterial potential as an antibiotic drug. Hence, we have identified twelve metabolites among our predictions, which according to different references, have antibacterial properties through either the direct or indirect inhibition of bacterial growth. For further convenience, we summarize our Journal validation results in Table 5.

#### 2.4.2. Structural Similarity

We successfully predicted 20 natural antibiotic compounds. We used 11 well-known antibiotics (Table 6) to validate our results based on structural similarity. Three types of similarity measures, Tanimoto, Dice, and Asymmetric, were employed to calculate the structural similarity between each pair of predicted and known antibiotics. A total of 20 × 11 = 220 similarity values were obtained for each type of measure.

The selection of the 11 antibiotics for comparison was based on their representation of diverse classes of antibiotics, such as β-lactams, fluoroquinolones, sulfonamides, rifamycins, and cephalosporins, each with distinct mechanisms of action, including the inhibition of cell wall synthesis, DNA replication, and folic acid synthesis. These antibiotics were extensively studied in clinical and experimental settings, resulting in a wealth of pharmacokinetic and pharmacodynamic data. Furthermore, comprehensive information on these antibiotics in DrugBank makes them suitable for similarity validation. By comparing predicted metabolites to these well-established antibiotics, this study aims to ensure that the novel compounds are benchmarked against clinically relevant and effective antibiotics, thereby enhancing the reliability of the results.

Similarity values greater than a threshold with known antibiotics were utilized to validate our predictions regarding ‘8-nitroguanosine 3′,5′-cyclic monophosphate’. Based on the Tanimoto similarity score, this metabolite exhibited high similarity to Rifaximin, with a value of 0.646. Utilizing Dice similarity, we found that our predicted metabolite shares similarities with several known antibiotics, including Daptomycin, Moxifloxacin, Rifaximin, Ciprofloxacin, Sulfamethoxazole, Trimethoprim, Amoxicillin, Cefdinir, Metronidazole, Cephalexin, and Levofloxacin. The highest similarity value achieved was 0.78. Additionally, we employed Asymmetric similarity to demonstrate that all our predicted metabolites share similarities with many known antibiotics, as shown in the heatmap of Figure 3. This heatmap indicates that all our predicted metabolites are structurally similar to at least one antibiotic with a similarity greater than 0.8.

## 3. Materials and Methods

The methods used in the current study are depicted in Figure 4. The primary steps involve (1) collecting data, (2) pre-processing the data, (3) creating a machine learning model, (4) ranking and selecting features, (5) validating the model, and (6) predicting metabolites.

### 3.1. Collecting Data

The original dataset obtained from [9] comprises 609 Unani formulas involving 369 herbal plants, categorized into 18 efficacy groups. The efficacy groups represent the types of diseases that Unani formulas can treat, including (1) Blood and Lymph Diseases, (2) Cancers, (3) Diseases of the Digestive System, (4) Ear, Nose, and Throat, (5) Diseases of the Eye, (6) Female-Specific Diseases, (7) Glands and Hormones, (8) The Heart and Blood Vessels, (9) Diseases of the Immune System, (10) Male-Specific Diseases, (11) Muscle and Bone, (12) Neonatal Diseases, (13) The Nervous System, (14) Nutritional and Metabolic Diseases, (15) Respiratory Diseases, (16) Skin and Connective Tissue, (17) The Urinary System, and (18) Mental and behavioral disorders. These data were obtained from the book “BANGLADESH: National Formulary of Unani Medicine” (ISBN 978-984-33-3253-0). We utilized the same initial dataset as in our prior publication on Unani Medicine [9,15]. However, for our current research, we considered further detailed actions and applications of the formulas to divide them into two groups, antibiotic and non-antibiotic. In this case, diseases within each efficacy group may belong to either antibiotic or non-antibiotic group.

### 3.2. Data Pre-Processing

The initial Unani formulas consist of plant-based ingredients, and the Unani metabolites were obtained from various databases, including KNApSAcK Family Databases (http://www.knapsackfamily.com/KNApSAcK_Family, accessed on 12 December 2023), IJAH Analytics (http://ijah.apps.cs.ipb.ac.id, accessed on 12 December 2023), KEGG (https://www.genome.jp/kegg/, accessed on 12 December 2023), and ChEBI (https://www.ebi.ac.uk/chebi/, accessed on 12 December 2023). The number of metabolites in each herbal plant varies greatly, with some plants having only a few metabolites while others have many. The KNApSAcK database contains information on the species–metabolite relationship, including accurate mass, molecular formula, metabolite name, and mass spectra in several ionization modes. It may contain common metabolites between Jamu and Unani, as both are classified as traditional medicine. IJAH Analytics is an open-access database specifically for Jamu data, providing plant–metabolite relations. KEGG is another open-access database containing cell, organism, and molecular information and large-scale molecular datasets. ChEBI is a database containing molecular entities, primarily focusing on small chemical compounds.

The next step in the process is to assign the formulas into two categories: antibiotics and non-antibiotics. The (class 1) “antibiotics” indicate that the corresponding Unani formula can cure diseases caused by bacteria. In contrast, the (class 0) “non-antibiotics” signifies that the corresponding Unani formula can cure diseases except those caused by bacteria 2. Our colleague, who has a medical background, assisted in this process. For example, the formula called “ArqSoya” that can cure Acidity of stomach and dysentery was assigned to antibiotic class. Contrary to that one, the formula called “Dawaul Misk Mutadlil” that can cure Weakness of brain, Heart and liver, Melancholia, and Insanity was assigned to non-antibiotics class. As a result, the final dataset is a tabular dataset consisting of Unani formulas as objects, metabolites as feature data, class labels, and the shape of our dataset is [381 × 4689]. The ratio of two classes in our data can be seen in Figure 5. A dataset in which the blue class (class 0) accounts for 68.8% of the samples and the orange class (class 1) represents 31.2% of the samples is considered imbalanced. This dataset’s class imbalance ratio (CIR) is 2.21, greater than 1. This ratio indicates more than twice as many samples in the pink class as in the blue class. Generally, a dataset is considered imbalanced when the CIR is greater than 10. However, even a CIR of 2.21 can pose problems for machine learning algorithms not explicitly designed to handle imbalanced datasets. It is essential to address the imbalance status of a dataset before training a machine learning model. This action can be achieved by oversampling the minority class, under-sampling the majority class, or using a machine learning algorithm that is designed to handle imbalanced datasets.
(1)$$CIR=\;\frac{Number\;of\;samples\;in\;Class\;0}{Number\;of\;samples\;in\;Class\;1}$$

Addressing the imbalanced data issue and fat data problems (where the number of features exceeds the number of instances), we have explored various pre-processing techniques, including adding synthetic data instances and feature selection methods. This research used SMOTE, PCA, and feature selection techniques to enhance the machine learning model’s performance.

### 3.3. Machine Learning Modeling

To determine the most appropriate machine learning method for our task, we thoroughly analyzed 12 approaches by applying them to the original data and optimizing the necessary parameters using grid search to ensure we attained the optimal results. The following list of classifiers was considered: AdaBoost, Bagging, BernoulliNB, Decision Tree, Extra Trees, Gradient Boosting, K-nearest neighbors, Linear Discriminant Analysis, Logistic Regression, Multilayer Perceptron, Random Forest, and Support Vector Machine. We summarize the machine learning modeling (Table 7).

### 3.4. Ranking and Selecting Feature

After finding the optimal pair of a machine learning model and pre-processing method for Unani-metabolite data, we utilized the variable importance attribute of the best model, which was implemented using the KerasRegressor and PermutationImportance package. We then attempted to rank the list of metabolites based on their weight (distribution of weight values can be seen in Figure 2), representing each feature’s importance for the network’s computations. These weights are adjusted during training to minimize the error between the network’s predictions and the target weight, meaning that the corresponding feature substantially impacts the network’s output.

KerasRegressor, which allows for the use of Keras models with the scikit-learn API, can be effectively combined with the PermutationImportance method for feature selection. After training a KerasRegressor model, the PermutationImportance method can be applied to assess the importance of each feature by evaluating the drop in model performance when the values of a feature are randomly shuffled. This approach is ideal for understanding the influence of individual input features on model predictions, particularly in complex neural networks. It is essential for enhancing interpretability and gaining insights into the importance of features for model accuracy and performance.

### 3.5. Validation

We used two methods to validate the predicted metabolites as natural antibiotics. First, we traced them directly to scientific journals or articles that reported their effectiveness in inhibiting bacterial growth; the summary of literature validation can be seen in Table 5. Second, we measured the structural similarity of the predicted metabolites with antibiotic compounds in the DrugBank database (https://go.drugbank.com/, accessed on 12 December 2023). We performed three similarity experiments: Tanimoto, Dice, and Asymmetric.

Journal validation entailed a comprehensive review of scientific journals and peer-reviewed articles that specifically documented the efficacy of natural compounds and metabolites in inhibiting bacterial growth, demonstrating antibacterial activity. This validation process was employed to ensure that our methodology aligns with established approaches in the field.

Structural similarity validation is a process in which we calculate the structural similarity of our metabolite prediction to 11 known antibiotics (The information about 11 antibiotics can be seen in Table 6). Metabolites with structures like antibiotics can enhance antibiotics’ bactericidal properties through various mechanisms, including increased antibiotic uptake, metabolic reprogramming, and induction of oxidative stress. e.g., metabolites such as glutamate, alanine, and glucose can potentiate the efficacy of antibiotics by enhancing their uptake and action within bacterial cells [46,47,48].

Tanimoto Similarity is a statistical parameter that quantifies the similarity between two sets of fingerprint bits typically represented in a binary form. It is widely used in cheminformatics to compute molecular similarities [49]. It was calculated as a ratio of the intersection of the two sets to the union of the two sets. The formula for the Tanimoto coefficient is defined as c/(a + b + c), where c represents the number of features standard to both compounds, a represents the number of features unique to one compound, and b represents the number of features unique to another compound. The Tanimoto coefficient ranges from 0 to 1, with higher values indicating more significant similarity between the two sets of fingerprint bits and lower values indicating lower similarity.

Dice similarity, or the Sørensen-Dice coefficient, is a statistical tool used to gauge the similarity between two sets. It benefits various fields, such as pattern recognition, medical diagnosis, and decision-making. The measure is defined as twice the size of the intersection of the sets divided by the sum of the sizes of the two sets.
(2)$$Dice\;Similarity\;\left(a,\;b\right)=\;\frac{2\times \left(Number\;of\;bits\;set\;in\;both\;fingerprint\right)\;}{Number\;of\;bits\;set\;in\;Fingerprint\;a+Number\;of\;bits\;set\;in\;Finger\;print\;b}$$

Asymmetric similarity is a metric utilized to evaluate the similarity between two molecules, accounting for distinctions in their structural features. The RDKit Python package offers a variety of similarity metrics, including asymmetric Tversky similarity, which is an extension of the Tanimoto similarity that permits unequal weighting of the features [50]. The formula for asymmetric Tversky similarity is as follows:(3)$$Asymmetric\;Tversky\;Similarity\;(a,\;b)=\frac{\Sigma (min(A[i],\;B[i]))+\alpha \times sum(min(A[i],\;B[i]))+\beta \times sum(max(A[i],\;B[i]))}{\Sigma (min(A[i],\;B[i]))+\alpha \times sum(min(B[i],\;A[i]))+\beta \times sum(max(A[i],\;B[i]))}$$
where:
*A* and *B* are the two vectors being compared, and alpha and beta are the weights assigned to the false positives and false negatives, respectively.*i* is the index of the feature being compared.

## 4. Conclusions

We developed a computational method to predict potential natural antibiotics using ingredients derived from Unani formulas. We utilized the Synthetic Minority Over-sampling Technique (SMOTE) and a Multi-Layer Perceptron (MLP) for classification, and our model demonstrated impressive performance with sensitivity, accuracy, and an F1-score reaching 91%, 83%, and 83.8%, respectively, in classifying Unani formulas based on their metabolite ingredients. Through our model, we identified 20 metabolites as potential antibiotics, and further analysis revealed that most had been previously reported to possess antibacterial properties in scientific journals. We compared the remaining metabolites to known antibiotics in the Drug Bank database to validate our predictions using similarity measures such as Tanimoto, Dice, and Asymmetric. Remarkably, these metabolites were structurally like 11 known antibiotics. These findings suggest that our computational approach holds promise for discovering new natural antibiotics, and the identified metabolites not only demonstrate antibacterial properties supported by the existing literature but also exhibit structural similarities to established antibiotics. Therefore, our findings provide a foundation for developing novel natural antibiotics with potential therapeutic applications.

## Figures and Tables

**Figure 1 antibiotics-13-00971-f001:**
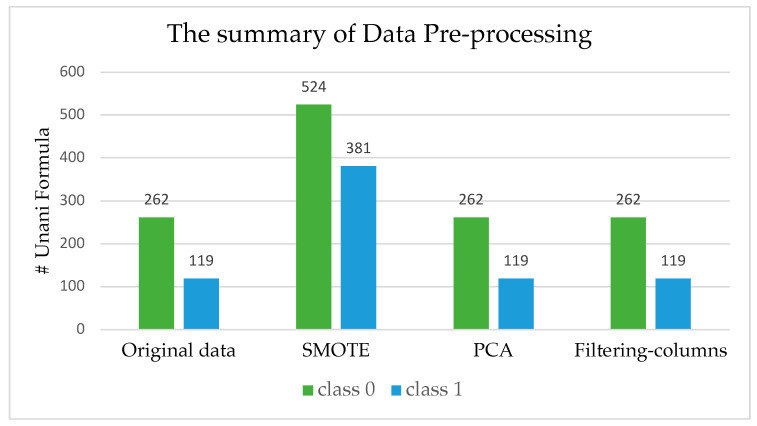
Summary of class label distribution of original data and after various pre-processing methods.

**Figure 2 antibiotics-13-00971-f003:**
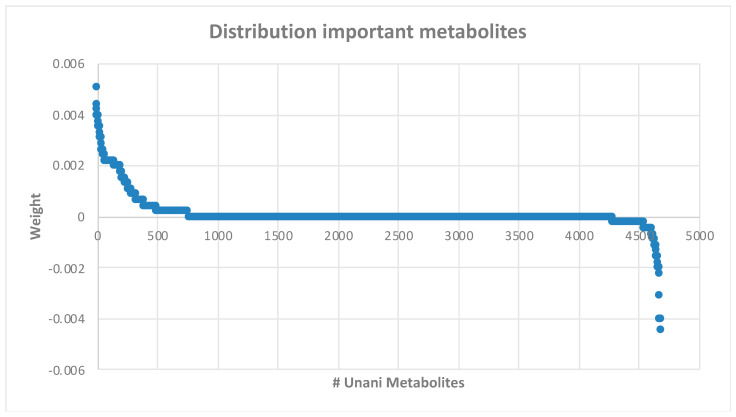
The weight distribution for all metabolites. The weight value indicates the importance of the variable in the MLP model. The higher the value, the more influential the feature is in making the model.

**Figure 3 antibiotics-13-00971-f004:**
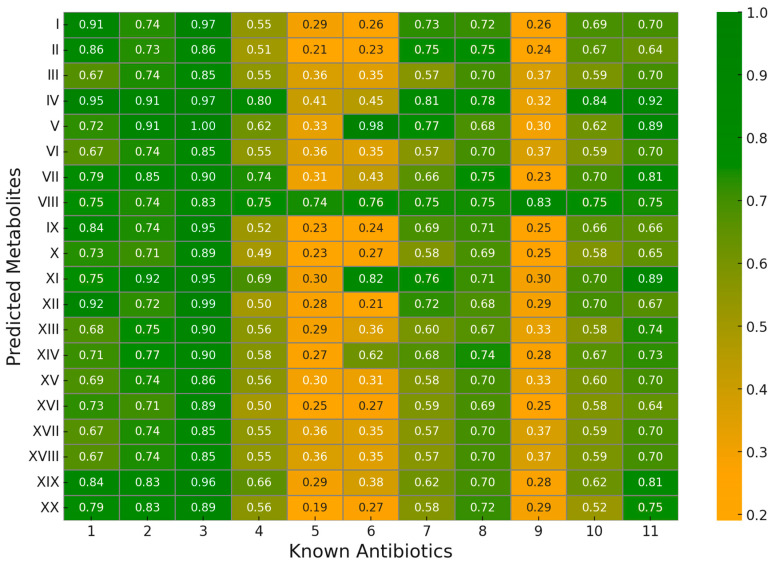
Asymmetric similarity results between predicted metabolites and known antibiotics. This heatmap presents the similarity scores between predicted metabolites (rows, the ID based on Table 4) and known antibiotics (columns, the ID based on Table 6). The color intensity represents the degree of similarity, with darker shades indicating higher similarity—variations in similarity scores across different antibiotic–metabolite pairs. For example, some antibiotics (e.g., column 3) show strong similarity with multiple metabolites, while others (e.g., column 6) exhibit lower similarity. The highest observed similarity is 1.00, while the lowest is 0.19.

**Figure 4 antibiotics-13-00971-f002:**
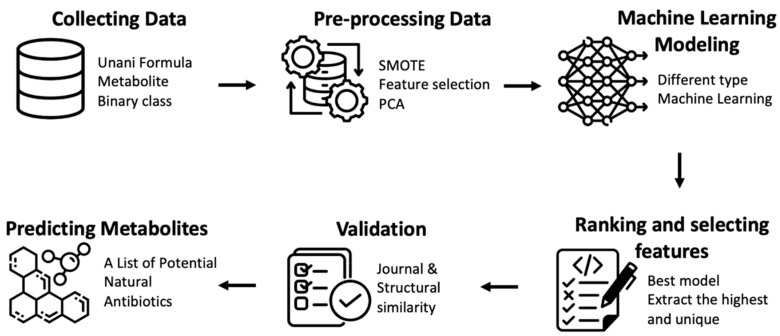
Methodology of research.

**Figure 5 antibiotics-13-00971-f005:**
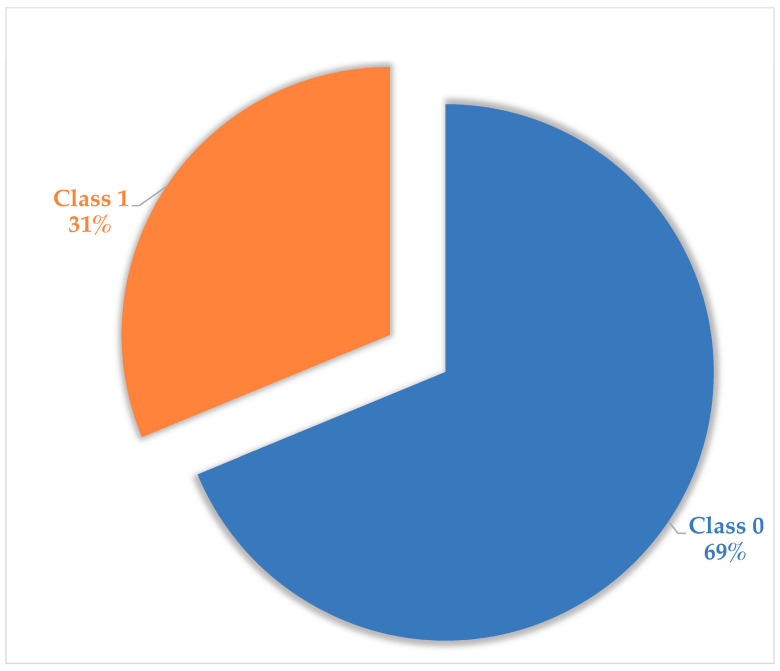
The ratio of two classes in the original dataset.

**Table 1 antibiotics-13-00971-t001:** Dataset summary before and after various pre-processing steps, including SMOTE, PCA, and feature selection.

Dataset	Number of Unani Formula	Number of Metabolite Features
Original data	381	4688
SMOTE	905	4688
PCA	381	21
Filtering columns	381	501

**Table 2 antibiotics-13-00971-t002:** Accuracy results of various machine learning models across different preprocessing techniques.

Machine Learning	Original Data	SMOTE	Feature Selection	PCA
AdaBoost	0.674596	0.742910	0.677193	0.681545
Bagging	0.697323	0.818048	0.698200	0.687685
BernouliNB	0.626361	0.614733	0.658715	0.615117
Decision Tree	0.666712	0.659669	0.680656	0.67979
Extra Trees	0.630793	0.828361	0.648348	0.632604
Gradient Boosting	0.685042	0.806262	0.685919	0.681556
K-Nearest Neighbors	0.688540	0.706446	0.688551	0.689417
Linier Discriminant Analysis	0.601003	0.750645	0.640351	0.664912
Logistic Regression	0.648257	0.764273	0.688551	0.662303
Multi-Layer Perceptron	0.694691	0.838200	0.694680	0.690305
Random Forest	0.687697	0.825046	0.691194	0.688596
Support Vector Machine	0.697323	0.776427	0.698200	0.691171

This table presents the accuracy scores of 12 machine learning algorithms, including AdaBoost, Bagging, Decision Tree, and Support Vector Machine, evaluated using original data and three preprocessed datasets: SMOTE, feature selection, and PCA. SMOTE generally improves accuracy across most models, particularly in Bagging (0.818048) and Multi-Layer Perceptron (0.838200), while feature selection and PCA have more mixed impacts depending on the model.

**Table 3 antibiotics-13-00971-t003:** Detailed performance metrics for the Multi-Layer Perceptron (MLP) model trained with the SMOTE dataset.

Measurement	Value	Derivation
Sensitivity	0.9182	TPR = TP/(TP + FN)
Specificity	0.7436	SPC = TN/(FP + TN)
Precision	0.7710	PPV = TP/(TP + FP)
Negative Predictive Value	0.9063	NPV = TN/(TN + FN)
False Positive Rate	0.2564	FPR = FP/(FP + TN)
False Discovery Rate	0.2290	FDR = FP/(FP + TP)
False Negative Rate	0.0818	FNR = FN/(FN + TP)
Accuracy	0.8382	ACC = (TP + TN)/(P + N)
F1-Score	0.8382	F1 = 2TP/(2TP + FP + FN)
Matthews Correlation Coefficient	0.6695	TP × TN − FP×FN/sqrt((TP + FP) × (TP + FN) × (TN + FP) × (TN + FN))

This table comprehensively evaluates the MLP model’s performance across multiple metrics, including sensitivity, specificity, precision, and accuracy. TP = true positive, TN = true negative, FP = false positive, FN = false negative.

**Table 4 antibiotics-13-00971-t004:** List of important metabolites predicted as natural antibiotics from the results of the best machine learning model.

Id	Metabolite	Formula	Molecular Mass (g/mol)
(I)	2-hydroxyethyl hexadecanoate	C_18_H_36_O_3_	300.48
(II)	N-[(+)-12-hydroxy-7 isojasmonyl]isoleucine	C_18_H_29_NO_5_	323.43
(III)	gluconapin	C_11_H_19_NO_9_S_2_	373.40
(IV)	3-phenylpropionitrile	C_9_H_9_N	131.18
(V)	flamenol	C_7_H_8_O_3_	140.14
(VI)	glucobrassicanapin(1−)	C_16_H_19_N_2_O_9_S_2_	446.46
(VII)	2-chlorobenzoic acid	C_7_H_5_ClO_2_	156.57
(VIII)	8-nitroguanosine 3′,5′-cyclic monophosphate	C_10_H_11_N_6_O_9_P	390.21
(IX)	1,3-dioctanoylglycerol	C_19_H_36_O_5_	344.49
(X)	PG(18:2(9Z,12Z)/16:0)	C_40_H_75_O_10_P	758.99
(XI)	3-methoxybenzaldehyde	C_8_H_8_O_2_	138.15
(XII)	Methyl stearate	C_19_H_38_O_2_	298.50
(XIII)	1-naphthyl β-D-glucoside	C_16_H_18_O_6_	306.32
(XIV)	sinapine	C_16_H_24_NO_5+_	310.36
(XV)	3-butenyldesulfoglucosinolate	C_12_H_23_NO_7_S	341.39
(XVI)	1-oleoyl-2-hexadecanoyl-sn-glycero-3-phospho-(1′-sn-glycerol-3′-phosphate)	C_18_H_34_O_2_	282.46
(XVII)	glucobrassicanapin	C_12_H_21_NO_9_S_2_	399.43
(XVIII)	gluconapin(1−)	C_11_H_18_NO_9_S_2_	373.40
(XIX)	5-methoxyindole-3-acetic acid	C_11_H_11_NO_3_	205.21
(XX)	α-allenylagmatine	C_8_H_16_N_4_	168.24

**Table 5 antibiotics-13-00971-t005:** Summary of the predicted metabolites.

Metabolite Name	Properties/Pharmacological Activities	Drug Target Protein	References
2-hydroxyethyl hexadecanoate (2-HEP)	Antibacterial; damages bacterial cell membranes and disrupts bacterial growth. Effective against *Staphylococcus aureus*, *Escherichia coli*, *Pseudomonas aeruginosa*.	Membrane phospholipids	[29]
Gluconapin	By myrosinase, and AITC has been demonstrated to be effective against various bacteria, including *Escherichia coli*, *Salmonella Typhimurium*, and *Staphylococcus aureus*	Membrane phospholipids	[30,31]
3-phenylpropionitrile	Antibacterial; disrupts bacterial membranes by forming pores. Effective against *Escherichia coli*, *Staphylococcus aureus*, *Pseudomonas aeruginosa*.	Bacterial cell membrane proteins (pore-forming)	[33]
Flamenol	Antibacterial properties against various bacteria, including *Staphylococcus aureus*, *Escherichia coli*, and *Pseudomonas aeruginosa*	-	[31]
Glucobrassicanapin(1−)	Antibacterial; a glucosinolate broken down into isothiocyanates, which have a broad range of antibacterial activities.	Enzyme involved in sulfur metabolism and membrane protein	[34]
2-chlorobenzoic acid (2-CBA)	Antibacterial; disrupts bacterial cell membranes and inhibits DNA synthesis. Effective against *Escherichia coli*, *Staphylococcus aureus*, *Bacillus subtilis*.	DNA replication proteins, membrane proteins	[35,36]
3-methoxybenzaldehyde	Antibacterial; disrupts bacterial cell membranes. Effective against Gram-positive and Gram-negative bacteria.	Membrane lipids and proteins	[37,38,39]
1-naphthyl β-D-glucoside	Antibacterial; inhibits bacterial glycosyltransferases, necessary for cell wall synthesis. Effective against multi-drug-resistant and non-resistant strains.	Cell wall synthesis	[40,41]
Sinapine	Antibacterial; disrupts bacterial cell membranes by interacting with phospholipids. Effective against antibiotic-resistant bacteria (e.g., MRSA).	Membrane phospholipids	[42,43]
3-butenyldesulfoglucosinolate	Antibacterial; derived from garlic, known for antimicrobial activity against bacteria such as *Staphylococcus aureus*.	Membrane proteins	[44,45]
Gluconapin(1−)	Antibacterial; converts into isothiocyanates, which disrupt bacterial cell membranes and inhibit enzymes. Effective against a broad spectrum of bacteria.	Membrane proteins	[31,32]
5-methoxyindole-3-acetic acid	Antibacterial; potential as an antioxidant and in the synthesis of agents. Limited information on antibacterial capabilities.	-	-

**Table 6 antibiotics-13-00971-t006:** Detailed information on antibiotics.

Id	Antibiotics	Formula	Molecular Mass (g/mol)	Rank
1	Daptomycin	C_72_H_101_N_17_O_26_	1620.69	3
2	Moxifloxacin	C_21_H_24_FN_3_O_4_	401.43	2
3	Rifaximin	C_43_H_51_N_3_O_11_	785.89	1
4	Ciprofloxacin	C_17_H_18_FN_3_O_3_	331.35	8
5	Sulfamethoxazole	C_10_H_11_N_3_O_3_S	253.28	11
6	Trimethoprim	C_14_H_18_N_4_O_3_	290.32	9
7	Amoxicillin	C_16_H_19_N_3_O_5_S	365.40	6
8	Cefdinir	C_14_H_13_N_5_O_5_S_2_	395.41	5
9	Metronidazole	C_6_H_9_N_3_O_3_	171.15	10
10	Cephalexin	C_16_H_17_N_3_O_4_S	347.39	7
11	Levofloxacin	C_18_H_20_FN_3_O_4_	361.37	4

The rank was determined based on the sum of the similarity scores between each known antibiotic and all the predicted antibiotics. The lowest rank corresponds to the highest sum of the scores.

**Table 7 antibiotics-13-00971-t007:** Summary of machine learning methods utilized in this work.

Machine Learning	Explanation	Advantage
AdaBoost	An ensemble learning algorithm that combines the predictions of multiple weak learners to produce a single, strong learner	Can handle imbalanced data
Bagging	Ensemble learning algorithm that works by creating multiple bootstrap samples of the training data and training a separate model on each sample	Can improve model with imbalanced data
BernoulliNB	Naive Bayes classifier that is specifically designed for binary classification problems	A simple but effective algorithm that is often used for imbalanced classification tasks
Decision Tree	Decision trees are a type of machine learning model that learns to classify data by constructing a tree of decision rules	Decision trees are relatively robust to imbalanced data, but they can be prone to overfitting
Extra Trees	Ensemble learning algorithm that is similar to random forests, but it uses a different approach to bootstrap sampling	Often used for imbalanced classification tasks because they are less likely to overfit than random forests
Gradient Boosting	An ensemble learning algorithm that combines the predictions of multiple weak learners in a sequential manner	A powerful algorithm that can be used for a variety of machine learning tasks, including imbalanced classification
K-nearest neighbors	K-nearest neighbors (KNN) is a simple but effective machine learning algorithm that classifies data by finding the K most similar training examples to a new data point and predicting the class of the new data point based on the classes of the K most similar training examples	Easy to implement and small dataset can use for imbalanced classification task
Linear Discriminant Analysis	Machine learning algorithm that projects the data onto a lower-dimensional space in a way that maximizes the discrimination between the different classes	Good choice for imbalanced data because it is able to find the most important features for discriminating between the classes
Logistic Regression	A machine learning algorithm that is used for binary classification problems	Often used for imbalanced classification task
Multilayer Perceptron	Multilayer perceptrons (MLPs) are a type of artificial neural network	Can use for classification task and able to learn complex data, even imbalanced classification task
Random Forest	Random forests are an ensemble learning algorithm that combines the predictions of multiple decision trees to produce a single prediction	Ability to handle imbalanced data and their resistance to overfitting
Support Vector Machine:	Machine learning that can use as classification and regression	Can handle high-dimensional data and imbalanced data

## Data Availability

We share data and code via GitHub (https://github.com/kamalNasution/Unani-Metabolite-Data-and-Code.git, uploaded on 7 October 2024).

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
