# Peer review of "Identifying Potential Natural Antibiotics from Unani Formulas through Machine Learning Approaches"

_antibiotics, 2024, doi:10.3390/antibiotics13100971_

Round 1

Reviewer 1 Report

Comments and Suggestions for Authors

Identifying potential antimicrobials from traditional formulars through modern technology is a promising alternative to tackle the antibiotic-resistant pathogens. The authors targeted Unani formulars for promising antimicrobials through 12 machine-learning approaches. It sounds good efforts for treatment of antibiotic-resistant pathogens. However, many challenges are arising, i.e. how to decide the formular candidates, how to identify molecules from these formulars, how to correlate these molecules with antimicrobial activity especially anti-antibiotics-resistant bacteria, how to treat dose-dependent or synergistic effects in the formulars and so on. It’s confusing of the principles applied in the identification of antimicrobials in this study. It is strongly recommended to provide a scheme to illustrate the roles of each machine learning in the prediction of antibiotics. If structural similarity with antibiotics is the final tool to validate the identified molecules, chemical library seems to be more competent than Unani formulars. The experimental test will be more persuasive to verify the efficacy of identified antibiotics.

36-48: Over detailed descriptions on VBNC were abrupt and misleading since the following work did not involve it.

358-369: The introduction of methodology was not informative enough, and what kinds of diseases were selected to classify these formulars to “antibiotics” or “non-antibiotics”? The diseases list was not specific enough to be attributed to bacteria-infection or physiological disorders. Even for the bacteria-infected diseases, they also could be caused by different bacteria. Were there any considerations on molecules with great efficacy to some special bacteria. I prefer to recommend a high-through antimicrobial Test to directly identify the antimicrobial activity of these formulars.

Generally, it’s hard to discover new antibiotics based on such a process, and the authors also failed to identify new antimicrobials.

Author Response

Reviewer 1

Reviewer’s comment:

Identifying potential antimicrobials from traditional formulars through modern technology is a promising alternative to tackle the antibiotic-resistant pathogens. The authors targeted Unani formulars for promising antimicrobials through 12 machine-learning approaches. It sounds good efforts for treatment of antibiotic-resistant pathogens. However, many challenges are arising, i.e. how to decide the formular candidates, how to identify molecules from these formulars, how to correlate these molecules with antimicrobial activity especially anti-antibiotics-resistant bacteria, how to treat dose-dependent or synergistic effects in the formulars and so on. It’s confusing of the principles applied in the identification of antimicrobials in this study. It is strongly recommended to provide a scheme to illustrate the roles of each machine learning in the prediction of antibiotics. If structural similarity with antibiotics is the final tool to validate the identified molecules, chemical library seems to be more competent than Unani formulars. The experimental test will be more persuasive to verify the efficacy of identified antibiotics.

Our reply:

We thank the reviewer and appreciate the comment. Honestly this is a computational work initially focusing on finding new natural antibiotics, some of which might be effective against drug resistant bacteria or can be used as part of multi-drug therapy for synergistic effects against drug resistant bacteria. Further research is needed for pinpointing the usage of our predicted antibiotics. Unani formulas consist of plants as ingredients, and we extended those data to Plant Vs. Metabolite matrix by collecting metabolite content data of respective plants from KNApSAcK, IJAH Analytics databases and other online sources. Our method focusses on bioinformatics side which is to reduce the search space. We applied 12 machine learning methods with the target to find the suitable method to achieve the best classification results because no single method performs well on all types of data. Compounds derived from Unani formula using our method are very likely to be effective antibiotics in the first position and second, we compared their structure with known antibiotics for further support. Because, in many cases the substructure of a drug is effective part and this comparison with Unani compounds might be meaningful instead of comparing with arbitrary compounds.

We thank our reviewer for his concerns. We added some of these sentences in the revised manuscript (Introduction and Validation sections) for better understanding of the readers.

Reviewer’s comment:

36-48: Over detailed descriptions on VBNC were abrupt and misleading since the following work did not involve it.

Our reply:

We appreciate your feedback, and we have revised the introduction section by omitting the VBNC description. The following is our revised introduction.

“Antibiotic resistance presents a primary global health concern involving the spreading of bacteria and genetic material among humans, animals, and the environment [1]. Antibiotic resistance can enhance bacteria's ability to withstand antibiotics and medications. Developing new antibiotics is challenging, time-consuming, and expensive, making this issue particularly alarming. Failure to address this problem promptly and effectively could result in an estimated 10 million deaths annually due to antibiotic resistance by 2050 [2]. According to the European Center for Disease Prevention and Control (ECDC), around 33,000 people die annually due to antibiotic-resistant issues. Epidemiologists emphasize the substantial economic consequences of antibiotic resistance, stating that in the United States and other countries, the additional hospitalizations and treatment costs due to superbugs or antibiotic-resistant problems exceed 11 million and $20 billion, respectively [3]. …… “

Reviewer’s comment:

358-369: The introduction of methodology was not informative enough, and what kinds of diseases were selected to classify these formulars to “antibiotics” or “non-antibiotics”? The diseases list was not specific enough to be attributed to bacteria-infection or physiological disorders. Even for the bacteria-infected diseases, they also could be caused by different bacteria. Were there any considerations on molecules with great efficacy to some special bacteria. I prefer to recommend a high-through antimicrobial Test to directly identify the antimicrobial activity of these formulars.

Our reply:

We greatly appreciate your detailed feedback regarding methodology, particularly how we classified diseases/formulas into two groups as "antibiotics" or "non-antibiotics."

As we mentioned in the methodology, we were helped by our colleague who has medical background. For example, the formula called “ArqSoya” that can cure Acidity of stomach and dysentery was assigned to antibiotic class. Contrary to that the formula called “Dawaul Misk Mutadlil” that can cure Weakness of brain, Heart and liver, Melancholi, and Insanity was assigned to non-antibiotics class.

Our original data was divided into to 18 efficacies for previous studies, but for our current research we considered further detail actions and applications of the formulas to divide them into two groups, antibiotic and non-antibiotic. In this case diseases within each efficacy group may belong to either antibiotic or non-antibiotic group. We added such information in the revised manuscript (Preprocessing section).

Reviewer 2 Report

Comments and Suggestions for Authors

The manuscript entitled “Identifying Potential Natural Antibiotics from Unani Formulas through Machine Learning Approaches” submitted by Ahmad Kamal Nasution et al to the journal -Antibiotics was reviewed. The manuscript has minor changes to be incorporated prior to its acceptance.

Self-explanatory legends can be incorporated for Figures. Especially figure 3 description in naive.

Table captions can be more descriptive, present form is very simple and chance for misguiding.

In table 5, additional column may be incorporated to represent the drug target protein information.

Why specifically these 11 antibiotics were used for comparison in this study. Brief statements can be incorporated to support this.

Microorganisms name should in italics and on second instance genus name can be abbreviated.

Reference can be included for this statement “Failure to address this problem promptly and effectively could result in an estimated 10 million deaths annually due to antibiotic resistance by 2050.”

Abbreviate and acronym the terms then and there throughout the manuscript. For Ex: PCA.

Author Response

Reviewer 2

Reviewer’s comment:

Self-explanatory legends can be incorporated for Figures. Especially figure 3 description in naive.
Our reply:

We thank the reviewer and appreciate the comment. We agree that the figure legends can be made more self-explanatory. We have updated the description for Figure 3 to enhance clarity and provide more context for the readers as follows:

Figure 3. Asymmetric similarity results between predicted metabolites and known antibiotics. This heatmap presents the similarity scores between predicted metabolites (rows) and known antibiotics (columns). The color intensity represents the degree of similarity, with darker shades indicating higher similarity—variations in similarity scores across different antibiotic-metabolite pairs. For example, some antibiotics (e.g., column 3) show strong similarity with multiple metabolites, while others (e.g., column 6) exhibit lower similarity. The highest observed similarity is 1.00, while the lowest is 0.19.

Also, we updated other figure legends with more information.

Reviewer’s comment:

Table captions can be more descriptive, present form is very simple and chance for misguiding.
Our reply:

We thank the reviewer and appreciate the comment. We added some explanations in captions of Table 1, Table 2, Table 3, Table 4, and Table 5 5 in the revised manuscript. We also somewhat modified the content of some tables based on other reviewer’s comment.

Reviewer’s comment:

In table 5, additional column may be incorporated to represent the drug target protein information.

Our reply:

We agree to add more information in Table 5. In the updated the manuscript head of Table 5 is as following.

Table 5. Summary of the predicted metabolites.

Metabolite Name

Properties/Pharmacological Activities

Drug Target Protein

References

2-hydroxyethyl hexadecanoate (2-HEP)

Antibacterial; damages bacterial cell membranes and disrupts bacterial growth. Effective against Staphylococcus aureus, Escherichia coli, Pseudomonas aeruginosa.

Membrane phospholipids

[18, 19, 20, 21]

3-phenylpropionitrile

Antibacterial; disrupts bacterial membranes by forming pores. Effective against Escherichia coli, Staphylococcus aureus, Pseudomonas aeruginosa.

Bacterial cell membrane proteins (pore-forming)

[20, 22]

Reviewer’s comment:

Why specifically these 11 antibiotics were used for comparison in this study. Brief statements can be incorporated to support this.

Our reply:

We appreciate the comment. We added following text in 2.4.2 section to address the reviewer’s suggestion.

“The selection of the 11 antibiotics for comparison was based on their representation of diverse classes of antibiotics, such as β-lactams, fluoroquinolones, sulfonamides, rifamycins, and cephalosporins, each with distinct mechanisms of action, including inhibition of cell wall synthesis, DNA replication, and folic acid synthesis. These antibiotics have been extensively studied in clinical and experimental settings, resulting in a wealth of pharmacokinetic and pharmacodynamic data. Furthermore, comprehensive information on these antibiotics in DrugBank makes them suitable for similarity validation. By comparing predicted metabolites to these well-established antibiotics, this study aims to ensure that the novel compounds are benchmarked against clinically relevant and effective antibiotics, thereby enhancing the reliability of the results”.

Reviewer’s comment:

Microorganisms name should in italics and on second instance genus name can be abbreviated.

Our reply:

Thank you for bringing this to our attention. We have carefully reviewed the manuscript and updated all microorganism names in italics.

Reviewer’s comment:

Reference can be included for this statement “Failure to address this problem promptly and effectively could result in an estimated 10 million deaths annually due to antibiotic resistance by 2050.

Our reply:

Thank you. We have updated our manuscript. The following is the updated sentence in the introduction section:

“…. Failure to address this problem promptly and effectively could result in an estimated 10 million deaths annually due to antibiotic resistance by 2050 [2]. ……”

Reviewer’s comment:

Abbreviate and acronym the terms then and there throughout the manuscript. For Ex: PCA.

Our reply:

We have carefully reviewed the manuscript and updated it:

e.g., the following is the part of the updated abstract:

“… We used 12 machine-learning algorithms and several techniques for preprocessing data, such as Synthetic Minority Over-sampling Technique (SMOTE), Feature Selection, and Principal Component Analysis (PCA)…. “   

Reviewer 3 Report

Comments and Suggestions for Authors

My comments (22 points): rich in good, novelty/innovative, methodology, combination traditional and ML/AI tools 

1, Abstract: some key essential index or parameters related to machine learning are advised to note as you can except protocol list/agenda of your work without core content;

2, Keywords: Unani medicine is OK, but any other popular/common word together is advised if possible, so that anyone knows its exact just one a look;

3,  Line 59-60: ...using various database sources such as drug repurposing hub and ZINC15,  a suitable reference is cited here if possible?

4, Line 60, ...eight antibiotic compounds, could they be listed one by one?

5, Line 61-62, Using eight learning methods, gram stain data, site of infection, and patient demographics..., some references was cited here probably? They are new faces or tools for most common readers perhaps?

6, Line 66-67, this paper employed machine learning and graph/network theory techniques, here some citations are necessary?

7, Line 74-75, ...and not less than 80% of people worldwide depend on herbal medicines. References necessary here? you give exact data, references are necessary accordingly?

8,  Line 77-79, The use of herbal medicines worldwide reached US$ 60 billion in 2010 and is expected to reach US$ 5 trillion by 2050  [14][15]. from a view of reader, a data around 2024 is hoped, you can check more or predict by youself or not? Interesting and important?

9,  Line 88, Researchers have yet to conduct much research on building Unani's scientific foundation, (1) your meaning right or OK? (2) references hoped also, because this is interesting with narrow path or area, more cotations are better for well undersanding? 

10, Line 91-99, whole of your five/six sentences as foolws: (1) The Unani System of Medicine was  invented in Greece and refined by Arabs into a sophisticated medical discipline using the ‘Hippocrates and Jalinoos’ teachings (Galen). (2) Unani medicine has since been referred to as Greco-Arab Medicine. (3) The Hippocratic notion of the four humors was blood, phlegm, 94 yellow bile, and black bile. (4) According to this approach, these principles govern the health 95 and composition of the body and its pathological states. (5) The Unani System of Medicine (USM) has been acknowledged by the World Health Organization (WHO) as an alternative system to meet the demands of the human population in terms of health care. (6) The practice of alternative medicine has become widespread.

One reference is for one sentence is better if possible, deending on you? 

11, Table 1, four/4 dataset, their www link information should be noted together here even though they are listed in Section Method if possible?

12, Figure 1, collumn color should be kept as united/ same color degree /level/density?

13, Table 2 , table 4 and table 7, and even Table 3 and Figure 2 and Figure 3, endnotes in more details should be enhanced in self description style with common explanation so that it can be directly easy understand if possible

14,  after Line 199, 2.4. Validation and Table 5, we hope to read out some exact conclusive sentences clearly after your vallidation instead of step/tool  description stackelling without any essentila context, i.e., any key data as threhold, parameter, entrance or other feature quanitative value or key structure or groups or special bonds and even possible target molecule or point like those in Table 5 except plant organ source and these general core group? so that they can dialogure with modern mecdicine, bridge meaning? as you can try it more if possible  

15, Table 6, Could some possibility or percentage data rank from high to low related to Unani medicine be predicted or determined if possible?

16, Fig 5, letters in Fig seem too small, largen them more?

17,  Section 3.4 and 3.5, long description should be matched with some key data as parameters, threhold and scope distribution from Figures and Tables and even ML or AI tools if possible?

18, Reference list, two 9? check and confirm them and others?

19, Refence 11, Methods, why typed as italic form? check and confirm through your all full text

20, Line 593, Allium sativum, it should be typed as italic form and check other possible similar errors? 

21, Check, confirm and united,some references are shown with only their first page, and some with strat and end page number, they should be united according to general principles in this journal.

22. Percent match: 26%,iThenticate report, it had better if lower.

(end at morning 12 Sept2024 at my office)   

Author Response

Reviewer 3

Reviewer’s comment:

1, Abstract: some key essential index or parameters related to machine learning are advised to note as you can except protocol list/agenda of your work without core content;.
Our reply:

We thank the reviewer and appreciate the comment. We made minor updates in our abstract. “Abstract: The Unani Tibb is a medical system of Greek descent that has undergone substantial dissemination since the 11th century and is currently prevalent in modern South and Central Asia, particularly in primary health care. The ingredients of Unani herbal medicines are primarily derived from plants. Our research aimed to address the pressing issues of antibiotic resistance, multi-drug resistance, and the emergence of superbugs by examining the molecular-level effects of Unani ingredients as potential new natural antibiotic candidates. We utilized a machine learning approach to tackle these challenges, employing decision trees, kernels, neural networks, and probability-based methods. We used 12 machine-learning algorithms and several techniques for preprocessing data, such as Synthetic Minority Over-sampling Technique (SMOTE), Feature Selection, and Principal Component Analysis (PCA). To ensure that our model was optimal, we conducted grid-search tuning to tune all the hyperparameters of the machine-learning models. The application of Multi-Layer Perceptron (MLP) with SMOTE pre-processing techniques resulted in impressive accuracy, precision and recall values. This analysis identified 20 important metabolites as essential components of the formula, which we predict as natural antibiotics. In the final stage of our investigation, we verified our prediction by conducting a literature search for journal validation or by analyzing the structural similarity with known antibiotics using asymmetric similarity.”

Reviewer’s comment:

2, Keywords: Unani medicine is OK, but any other popular/common word together is advised if possible, so that anyone knows its exact just one a look.
Our reply:

We appreciate the comment. We agreed and updated the keyword to “Unani herbal medicine”

Reviewer’s comment:

3,  Line 59-60: ...using various database sources such as drug repurposing hub and ZINC15,  a suitable reference is cited here if possible?.
Our reply:

We thank the reviewer and appreciate the comment. We have updated our manuscript as follows.

“… using various database sources such as drug repurposing hub [5] and ZINC15 [6] ….”

Reviewer’s comment:

4, Line 60, ...eight antibiotic compounds, could they be listed one by one?.
Our reply:

We thank the reviewer and appreciate the comment. We have updated our manuscript.

“… the eight antibiotics are ZINC000098210492, ZINC000001735150, ZINC000225434673, ZINC000004481415, ZINC000019771150, ZINC000004623615, ZINC000238901709, and ZINC000100032716. … “

Reviewer’s comment:

5, Line 61-62, Using eight learning methods, gram stain data, site of infection, and patient demographics..., some references was cited here probably? They are new faces or tools for most common readers perhaps?.
Our reply:

We thank the reviewer and appreciate the comment. We have added the reference to this sentence and revised it slightly.

“… Using eight machine learning methods, gram stain data, site of infection, and patient demographics were utilized to build decision tools for determining antimicrobial-resistance [7] … “

Reviewer’s comment:

6, Line 66-67, this paper employed machine learning and graph/network theory techniques, here some citations are necessary?
Our reply:

We thank the reviewer and appreciate the comment. We have updated the manuscript.

“… Other work utilized traditional Chinese medicine to seek potential natural products as antibiotics; this paper employed machine learning and graph/network theory techniques [8, 9]. … “

Reviewer’s comment:

7, Line 74-75, ...and not less than 80% of people worldwide depend on herbal medicines. References necessary here? you give exact data, references are necessary accordingly?.
Our reply:

We thank the reviewer and appreciate the comment. We have updated the manuscript.

“… Herbal medicine has become a popular drug in the last three decades, and not less than 80% of people worldwide depend on herbal medicines [11]. …”

Reviewer’s comment:

8, Line 77-79, The use of herbal medicines worldwide reached US$ 60 billion in 2010 and is expected to reach US$ 5 trillion by 2050  [14][15]. from a view of reader, a data around 2024 is hoped, you can check more or predict by youself or not? Interesting and important?.
Our reply:

We thank the reviewer and appreciate the comment. We could not find paper references for 2024, but we found the paper for 2016, so we added these references to our manuscript.

“… The use of herbal medicines worldwide reached US$ 60 billion in 2010, US$ 71.19 in 2016 and is expected to reach US$ 5 trillion by 2050 [12, 13, 14].  “

Reviewer’s comment:

9,  Line 88, Researchers have yet to conduct much research on building Unani's scientific foundation, (1) your meaning right or OK? (2) references hoped also, because this is interesting with narrow path or area, more cotations are better for well undersanding? .

10, Line 91-99, whole of your five/six sentences as foolws: (1) The Unani System of Medicine was  invented in Greece and refined by Arabs into a sophisticated medical discipline using the ‘Hippocrates and Jalinoos’ teachings (Galen). (2) Unani medicine has since been referred to as Greco-Arab Medicine. (3) The Hippocratic notion of the four humors was blood, phlegm, 94 yellow bile, and black bile. (4) According to this approach, these principles govern the health 95 and composition of the body and its pathological states. (5) The Unani System of Medicine (USM) has been acknowledged by the World Health Organization (WHO) as an alternative system to meet the demands of the human population in terms of health care. (6) The practice of alternative medicine has become widespread.

One reference is for one sentence is better if possible, deending on you? 
Our reply:

We thank the reviewer and appreciate the comment. We have updated our manuscript by adding as much references as possible.

“ … The Unani Tibb, known as Unani medicine, is widely practiced in South and Central Asia. The Arabic term "Tibb" means "medicine," while the name "Unani" is assumed to have its roots in the Greek word "Ionan" [15]. Traditional Indian and Chinese systems also influenced it. The Unani herbal medicines primarily utilize medicinal plants as their ingredients, and this system follows ancient concepts and principles of drug management. Researchers have yet to conduct much research on building Unani's scientific foundation. This is needed to provide a foundation and knowledge of why an Unani formula is helpful for a particular disease. Unani medicines are made by extracting medicinal plants used as drugs against various diseases [16]. Based on [17], The Unani System ….”

Reviewer’s comment:

11, Table 1, four/4 dataset, their www link information should be noted together here even though they are listed in Section Method if possible?.
Our reply:

We thank the reviewer and appreciate the comment. we first apologize, as this research is still ongoing, and we have decided to keep the original data confidential for now. However, should there be any special requests, readers are welcome to contact us directly, and we would be happy to share the data upon consideration.

Reviewer’s comment:

12, Figure 1, collumn color should be kept as united/ same color degree /level/density?
Our reply:

We thank the reviewer and appreciate the comment. We have updated in manuscript.

Figure 1. Summary of class label distribution of original data and after various pre-processing methods.

Reviewer’s comment:

13, Table 2 , table 4 and table 7, and even Table 3 and Figure 2 and Figure 3, endnotes in more details should be enhanced in self description style with common explanation so that it can be directly easy understand if possible.
Our reply:

We thank the reviewer and appreciate the comment. We added some explanations in the captions of Table 1, Table 2, Table 3, Table 4, and Table 5 in the revised manuscript. We also somewhat modified the content of some tables based on other reviewer’s comment. Here is an example of a caption in Figure 3.

“ … Figure 3. Asymmetric similarity results between predicted metabolites and known antibiotics. This heatmap presents the similarity scores between predicted metabolites (rows, the Id based on Table 4) and known antibiotics (columns, the Id based on Table 6). The color intensity represents the degree of similarity, with darker shades indicating higher similarity—variations in similarity scores across different antibiotic-metabolite pairs. For example, some antibiotics (e.g., column 3) show strong similarity with multiple metabolites, while others (e.g., column 6) exhibit lower similarity. The highest observed similarity is 1.00, while the lowest is 0.19. “

Reviewer’s comment:

14,  after Line 199, 2.4. Validation and Table 5, we hope to read out some exact conclusive sentences clearly after your vallidation instead of step/tool  description stackelling without any essentila context, i.e., any key data as threhold, parameter, entrance or other feature quanitative value or key structure or groups or special bonds and even possible target molecule or point like those in Table 5 except plant organ source and these general core group? so that they can dialogure with modern mecdicine, bridge meaning? as you can try it more if possible  .
Our reply:

We thank the reviewer and appreciate the comment. We agree to add more information in Table 5. The following is the head of the updated Table 5.

Table 5. Summary of the predicted metabolites.

Metabolite Name

Properties/Pharmacological Activities

Drug Target Protein

References

2-hydroxyethyl hexadecanoate (2-HEP)

Antibacterial; damages bacterial cell membranes and disrupts bacterial growth. Effective against Staphylococcus aureus, Escherichia coli, Pseudomonas aeruginosa.

Membrane phospholipids

[18, 19, 20, 21]

Reviewer’s comment:

15, Table 6, Could some possibility or percentage data rank from high to low related to Unani medicine be predicted or determined if possible?
Our reply:

We thank the reviewer and appreciate the comment. We agree to add a column to enrich our paper. Here is the Table 6

Table 6. Details information on antibiotics.

Id

Antibiotics

Formula

Molecular Mass (g/mol)

Rank

1

Daptomycin

C72H101N17O26

1620.69

3

2

Moxifloxacin

C21H24FN3O4

401.43

2

3

Rifaximin

C43H51N3O11

785.89

1

4

Ciprofloxacin

C17H18FN3O3

331.35

8

5

Sulfamethoxazole

C10H11N3O3S

253.28

11

6

Trimethoprim

C14H18N4O3

290.32

9

7

Amoxicillin

C16H19N3O5S

365.40

6

8

Cefdinir

C14H13N5O5S2

395.41

5

9

Metronidazole

C6H9N3O3

171.15

10

10

Cephalexin

C16H17N3O4S

347.39

7

11

Levofloxacin

C18H20FN3O4

361.37

4

* The rank was determined based on the sum of the similarity scores between each known antibiotic and all the predicted antibiotics. The lowest rank corresponds to the highest sum of the scores.

Reviewer’s comment:

16, Fig 5, letters in Fig seem too small, largen them more?
Our reply:

We thank the reviewer and appreciate the comment. We have updated in manuscript.

Figure 5. The ratio of two classes in the original dataset.

Reviewer’s comment:

17,  Section 3.4 and 3.5, long description should be matched with some key data as parameters, threhold and scope distribution from Figures and Tables and even ML or AI tools if possible?

Our reply:

We thank the reviewer and appreciate the comment. We have updated the manuscript by referring to Figure 2, Table 5, and Table 6.

Reviewer’s comment:

18, Reference list, two 9? check and confirm them and others?
Our reply:

We thank the reviewer and appreciate the comment. We have revised our manuscript.

“….

  1. Gao, P.; Nasution, A.K.; Yang, S.; Chen, Z.; Ono, N.; Kanaya, S.; Altaf-Ul-Amin, M.D. On finding natural antibiotics based on TCM formulae. Methods 2023, 214, 35–45.
  2. Nasution, A.K.; Ono, N.; Kanaya, S.; Ul-Amin, M.A. Investigating Potential Natural Antibiotics Plants Based on Unani Formula Using Supervised Network Analysis and Machine Learning Approach. In Proceedings of the 2023 IEEE International Conference on Bioinformatics and Biomedicine (BIBM), Istanbul, Turkey, 3–5 December 2023; IEEE: New York, NY, USA, 2023; pp. 3111–3117.

….“

Reviewer’s comment:

19, Refence 11, Methods, why typed as italic form? check and confirm through your all full text
Our reply:

We thank the reviewer and appreciate the comment. We have revised our manuscript.

Reviewer’s comment:

20, Line 593, Allium sativum, it should be typed as italic form and check other possible similar errors?
Our reply:

Thank you for bringing this to our attention. We have carefully reviewed the manuscript and updated all scientific species names in italics.

Reviewer’s comment:

21, Check, confirm and united,some references are shown with only their first page, and some with strat and end page number, they should be united according to general principles in this journal.
Our reply:

We thank the reviewer and appreciate the comment. We have carefully reviewed the manuscript and updated the missing reference format. However, some of the references we could not find the page range on the website.

Reviewer’s comment:

  1. Percent match: 26%,iThenticate report, it had better if lower.

Our reply:

We thank the reviewer and appreciate the comment. We hope it would be lower in the revised manuscript.

Reviewer 4 Report

Comments and Suggestions for Authors

The authors present an in silico analysis of compounds found in treatments in the traditional unani system for antimicrobial activity.  While the mining of natural products and traditional medicines is a HIGHLY valuable pursuit, there are a number of serious gaps in the presented work that must be addressed before publication.

1.  The most glaring issue with the manuscript is the relative disconnect between the standard types of articles in the journal and this one.  The content is 100% relevant, however the authors do not make a strong enough connection between the system (i.e. the Unani traditional medicine approaches) and the antimicrobial activity from these compounds/sources.  Suggestions to improve this aspect:

(A) The authors must expand the introduction of the unani system.  What sources do these compounds come from in the system?  While the authors clearly state that Unani herbal medicines are under-studied, the readers should be able to do further research if the plants/sources are identified.  

(B) it is ESSENTIAL that the antibacterial activity of the compounds be more clearly presented.  The weight analysis is confusing to experimentalists, as this could refer to weight % in the natural product source, MW, etc.  The Table 5 should include references, and add a column indicating the strains the compound was shown to be active against.

(C) Figure 3 is highly valuable!  However it is un-interpretable because the predicted metabolites are not labeled/referenced/identified!  

(D) Tables 4 and 6 need more information.  The simple atomic composition of the compounds is insufficient to perform any serious interpretation or comparison of the molecules in question.  Minimally structures of those compounds should be included.

2. overall the figure legends are lacking sufficient detail to interpret the figure. I would suggest reviewing examples for figure legends from published papers in the journal and expanding.

3.  Figure 1 should be remade/revised without the fading effect

4. Section 2.4.1 should be retitled "literature validation"

5.  Figure 5 - the text in the figure is not legible...too small

6.  there were numerous instances throughout the manuscript where bacterial names are not italicized

Comments on the Quality of English Language

fine

Author Response

Reviewer 4

Reviewer’s comment:

  1. The most glaring issue with the manuscript is the relative disconnect between the standard types of articles in the journal and this one.  The content is 100% relevant, however the authors do not make a strong enough connection between the system (i.e. the Unani traditional medicine approaches) and the antimicrobial activity from these compounds/sources.  Suggestions to improve this aspect:

(A) The authors must expand the introduction of the unani system.  What sources do these compounds come from in the system?  While the authors clearly state that Unani herbal medicines are under-studied, the readers should be able to do further research if the plants/sources are identified.  

(B) it is ESSENTIAL that the antibacterial activity of the compounds be more clearly presented.  The weight analysis is confusing to experimentalists, as this could refer to weight % in the natural product source, MW, etc.  The Table 5 should include references, and add a column indicating the strains the compound was shown to be active against.

Our reply:

We thank the reviewer and appreciate the comment.

we classified Unani diseases/formulas into two groups as "antibiotics" or "non-antibiotics." In this task, we were helped by our colleague who has medical background. For example, the formula called “ArqSoya” that can cure Acidity of stomach and dysentery was assigned to antibiotic class. Contrary to that the formula called “Dawaul Misk Mutadlil” that can cure Weakness of brain, Heart and liver, Melancholi, and Insanity was assigned to non-antibiotics class.

  • Unani formulas are consisting of plants as ingredients and we extended those data to Plant Vs. Metabolite matrix by collecting metabolite content data of respective plants from KNApSAcK, IJAH Analytics databases and other online sources. We added this information in the revised manuscript. Furthermore, we provided predictions based on computational bioinformatics research, which are strong drug/antibiotic candidates for development by pharmacists or other related researchers.
  • The weight analysis in this paper does not relate to % in the natural product. It indicates the importance of a feature in the classification process of a machine learning model, based on which we decide which metabolites are more important predictions. To address this issue, we try to add more explanation in the legend of Figure 2.

“ … Figure 2. The weight distribution for all metabolites. The weight value indicates the importance of the variable in the MLP model. The higher the value, the more influential is the feature in making the model. …”

Reviewer’s comment:

(C) Figure 3 is highly valuable!  However it is un-interpretable because the predicted metabolites are not labeled/referenced/identified!  
Our reply:

We thank the reviewer and appreciate the comment. We have updated our manuscript related to Figure 3 and added more explanation in the Figure legend.

Figure 3. Asymmetric similarity results between predicted metabolites and known antibiotics. This heatmap presents the similarity scores between predicted metabolites (rows, the Id based on Table 4) and known antibiotics (columns, the Id based on Table 6). The color intensity represents the degree of similarity, with darker shades indicating higher similarity—variations in similarity scores across different antibiotic-metabolite pairs. For example, some antibiotics (e.g., column 3) show strong similarity with multiple metabolites, while others (e.g., column 6) exhibit lower similarity. The highest observed similarity is 1.00, while the lowest is 0.19

Reviewer’s comment:

(D) Tables 4 and 6 need more information.  The simple atomic composition of the compounds is insufficient to perform any serious interpretation or comparison of the molecules in question.  Minimally structures of those compounds should be included.
Our reply:

We thank the reviewer and appreciate the comment. We have revised and added other information. In Table 4, we added the molecular mass. In Table 6, we add the molecular mass and rank (The rank was determined based on the sum of the similarity scores between each known antibiotic and all the predicted antibiotics. The lowest rank corresponds to the highest sum of the scores.). Please kindly see the update in our manuscript.

Reviewer’s comment:

  1. overall the figure legends are lacking sufficient detail to interpret the figure. I would suggest reviewing examples for figure legends from published papers in the journal and expanding.
  2. Figure 1 should be remade/revised without the fading effect
    Our reply:

We thank the reviewer and appreciate the comment. We have updated the manuscript.

Figure 1. Summary of class label distribution of original data and after various pre-processing methods.

Reviewer’s comment:

  1. Section 2.4.1 should be retitled "literature validation"
    Our reply:

We thank the reviewer and appreciate the comment. We agree and we updated.

“ …

2.4.1. Literature validation

We found supporting evidence for 10 of 20 of our predicted antibiotics in published literature. Below, we discuss this in detail.

….”

Reviewer’s comment:

  1. Figure 5 - the text in the figure is not legible...too small
    Our reply:

We thank the reviewer and appreciate the comment. We have updated the figure 5.

Figure 5. The ratio of two classes in the original dataset.

Reviewer’s comment:

  1. there were numerous instances throughout the manuscript where bacterial names are not italicized.

Our reply:

Thank you for bringing this to our attention. We have carefully reviewed the manuscript and updated all microorganism names in italics.

Round 2

Reviewer 1 Report

Comments and Suggestions for Authors

Most of my concerns have been well addressed, and I recommend its acceptance for publication in our journal.

Author Response

Reviewer 1

Round 2 

Reviewer’s comment:

Comments and Suggestions for Authors

Most of my concerns have been well addressed, and I recommend its acceptance for publication in our journal.

Our reply:

Thank you for your positive feedback and recommendation for publication. We appreciate your thoughtful review and are glad our revisions have addressed your concerns. We look forward to the next steps in the publication process.

Reviewer 3 Report

Comments and Suggestions for Authors Some more minor questions are still suspending as follows: 1, Line 63-64, ..., and not less than 80% of people worldwide..., I am not sure "not less than" should be repleced by "near" or "close to" or better, you can think and decide it. 2, Endnote in all of each Table and Figure should be separated from thier title, and put at end of bottom or below of each Table and each Figure instead of together with title. You can check or refer many papers do so as example. 3, Fig 1 and Fig 5 are not clear, their resolution ratio should improved more. 4, Then,I will recommend it for acceptance. (End at 27 Sept 2024) 

Author Response

Reviewer 3

Round 2

Reviewer’s comment:

Some more minor questions are still suspending as follows: 1, Line 63-64, ..., and not less than 80% of people worldwide..., I am not sure "not less than" should be repleced by "near" or "close to" or better, you can think and decide it. 

2, Endnote in all of each Table and Figure should be separated from thier title, and put at end of bottom or below of each Table and each Figure instead of together with title. You can check or refer many papers do so as example. 

3, Fig 1 and Fig 5 are not clear, their resolution ratio should improved more.

 4, Then,I will recommend it for acceptance. (End at 27 Sept 2024) 

Our reply:

  1. Thank you for pointing this out. After considering your suggestion, we agree that “close to 80%” sounds more natural. We have updated the sentences accordingly.
  2. Thank you for the comment, we have updated manuscript with separate endnotes for Table 2 and Table 3, we move the explanation to the bottom of the table as the note.  
  3. Thank you for your valuable feedback. We have addressed the clarity issue with Figures 1 and 5. We have improved the resolution of both figures by replacing them with vectorized versions, ensuring they maintain high clarity and sharpness at all viewing scales. Additionally, we have uploaded these vectorized figures to the manuscript revision website for further editing.
  4. Thank you for recommending for acceptance.

Reviewer 4 Report

Comments and Suggestions for Authors

significantly improved.  I think there is definitely room for more introduction/background on the unani plants and their applications, but perhaps the authors feel it is beyond the scope of this review.

Otherwise fine.

Author Response

Reviewer 4

Round 2

Reviewer’s comment:

significantly improved.  I think there is definitely room for more introduction/background on the unani plants and their applications, but perhaps the authors feel it is beyond the scope of this review.

Our reply:

Thank you for your suggestion. We have added some more information on Unani plants and their applications in our introduction as follows:

“… Unani medicines are utilized to treat versatile types of diseases. overview of the use of Unani plants in prevention and management of urolithiasis is discussed in [18]. Comprehensive discussion on Unani medicine system can be found in [19]. In our previous work we identified many plants used in Unani formulas as antibacterial [9]. For example, Piper longum (Pippali) has been traditionally used for various diseases such as asthma, insomnia, and diabetes and has been found to possess antibacterial activity against both Gram-positive and Gram-negative bacteria [20, 21]. Similarly, Trachyspermum ammi (Ajwain) essential oil has demonstrated potent antibacterial effects against pathogens such as Staphylococcus aureus, Pseudomonas aeruginosa, and Escherichia coli, highlighting its potential as a natural antibiotic [22]. Santalum album (Indian sandalwood), long recognized for its broad medicinal use, exhibits antimicrobial activity [23]. Cyperus rotundus essential oil has shown potent activity against Staphylococcus aureus [24]. Additionally, Vitis vinifera (grape) seed extracts have demonstrated significant antibacterial activity, particularly against Staphylococcus aureus, positioning it as a promising natural antibacterial agent [25, 26]. Matricaria chamomilla (chamomile) is well-known for its pharmacological properties, with its essential oil exhibiting notable antibacterial effects against various bacterial strains [27]. Lastly, Zingiber officinale (ginger) has been traditionally used for its antimicrobial properties, with both aqueous and alcoholic extracts showing efficacy against several bacterial strains [28]. These plants underscore the valuable role of natural products in Unani medicine for addressing bacterial infections. …”